# Environmentally Relevant Concentrations of Bisphenol A Interact with Doxorubicin Transcriptional Effects in Human Cell Lines

**DOI:** 10.3390/toxics7030043

**Published:** 2019-08-29

**Authors:** Edna Ribeiro, Mariana Delgadinho, Miguel Brito

**Affiliations:** H&TRC—Health & Technology Research Center, ESTeSL—Escola Superior de Tecnologia da Saúde, Instituto Politécnico de Lisboa, Av. D. João II, lote 4.69.01, Parque das Nações, 1990-096 Lisbon, Portugal

**Keywords:** Bisphenol A, doxorubicin, Hep-2 cell line, MRC-5 cell line, drug interaction, gene transcription

## Abstract

The worldwide production of synthetic chemicals, including endocrine disruptor chemicals (EDCs), such as Bisphenol A (BPA) has increased significantly in the last two decades. Human exposure to BPA, particularly through ingestion, is continuous and ubiquitous. Although, considered a weak environmental estrogen, BPA can induce divergent biological responses through several signaling pathways, including carcinogenesis in hormone-responsive organs. However, and despite the continuous increase of tumor cell-resistance to therapeutic drugs, such as doxorubicin (DOX), information regarding BPA drug interactions is still scarce, although its potential role in chemo-resistance has been suggested. This study aims to assess the potential interactions between environmentally relevant levels of BPA and DOX at a therapeutic dosage on Hep-2 and MRC-5 cell lines transciptome. Transcriptional effects in key-player genes for cancer biology, namely *c-fos*, *p21*, and *bcl-xl*, were evaluated through qRT-PCR. The cellular response was analyzed after exposure to BPA, DOX, or co-exposure to both chemicals. Transcriptional analysis showed that BPA exposure induces upregulation of *bcl-xl* and endorses an antagonistic non-monotonic response on DOX transcriptional effects. Moreover, the BPA interaction with DOX on *c-fos* and *p21* expression emphasize its cellular specificity and divergent effects. Overall, Hep-2 was more susceptible to BPA effects in a dose-dependent manner while MRC-5 transcriptional levels endorsed a non-monotonic response. Our data indicate that BPA environmental exposure may influence chemotherapy outcomes, which emphasize the urgency for a better understanding of BPA interactions with chemotherapeutic agents, in the context of risk assessment.

## 1. Introduction

The worldwide production of synthetic chemicals, such as endocrine disruptor chemicals (EDCs), has grown exponentially in the last two decades. Among EDCs, which are classified by the World Health Organization (WHO) as *“an exogenous substance or mixture that alters function(s) of the endocrine system, and consequently causes adverse health effects in an intact organism, or its progeny, or (sub) populations”* (World Health Organization 2011) [1,2], Bisphenol A (BPA) appears as a paradigmatic case. Currently, human exposure to this compound is persistent and continuous, particularly through ingestion of contaminated food and beverages. BPA also appears to have bioaccumulation capacity, which raises public and scientific concerns regarding its possible health-related effects [3]. Although, BPA is still the main identified bisphenol in environmental monitoring studies worldwide, diverse bisphenol analogues, such as Bisphenol F (BPF) and Bisphenol S (BPS), have been developed and produced, with the aim of replacing BPA in multiple applications [4]. Nevertheless, all these compounds have the molecular scaffold of bisphenols (BPA, bisphenol analogues, and bisphenol derivates) which result in the ability to interact with specific protein families, such as GTPases, including H-Ras, N-Ras, and K-Ras isoforms, crucial for cancer etiology [5,6].

Relevantly, the associations between human exposure to these compounds and hazardous effects on fertility, fetal development, and several types of cancer have been reported [7,8]. Even though BPA is considered a weak xenoestrogen, due to its low binding affinity to classical estrogen receptors (ERα and ERβ), this compound is capable of triggering diverse and divergent estrogen-signaling pathways [9,10], and to endorse tissue-specific, as well as developmental effects, including cellular proliferation and gene expression [11,12,13,14]. BPA-induced alterations on transcriptional patterns have been correlated to epigenetic effects, which were first demonstrated in mice after maternal exposure to BPA [15]. For the past two decades, BPA exposure effects have been exceedingly studied in carcinogenesis in hormone-responsive organs [16]. However, and despite the worldwide increase in cancer chemo-resistance, ref [17] information regarding potential interactions with chemotherapeutic drugs, such as Doxorubicin (DOX), has been scarce. DOX is one of the most utilized antineoplastic drugs for cancer treatments worldwide, ref [18] which at therapeutic concentrations, induces cancer cells apoptosis as a result of DNA damage [19]. Previous studies have demonstrated BPA’s counteracting effects in breast cancer cells [20,21] and HT29 cells [22] pre-exposed to DOX, which suggests an effective interaction and raises awareness regarding BPA’s role in chemo-resistance.

Moreover, considering that the ingestion of contaminated food and beverages are responsible for more than 90% of overall human environmental exposure to BPA at all age groups [23], the cells from the upper gastrointestinal tract, such as laryngeal cells, are particularly exposed to non-conjugated BPA (active form).

Here, we aimed to evaluate the effects of environmentally relevant concentrations of BPA, commonly found in human biological samples [24,25], and in co-exposure with a therapeutic concentration of DOX in human epithelial Type 2 (Hep-2) cells (originating from a human laryngeal carcinoma) and MRC-5 cells (DNA repair-proficient cell line). Our group’s previously published results, regarding BPA/DOX interactions in cytotoxicity and genotoxicity, as well as cytological evaluation of mitotic disruption, demonstrated its genotoxic potential, as well as antagonistic effect to DOX [26].

Here, the effects resulting from the combined exposures to BPA and DOX, were evaluated regarding the transcriptional disruption of genes, known as key players in cancer biology. The transcriptional effects were assessed for the anti-apoptotic gene *bcl-xl*, for which upregulation has been previously observed in BPA/DOX interaction [20]. While, *p21* gene, which encodes for a cyclin-dependent kinase inhibitor, is particularly related to cell cycle progression and cancer outcome [27] and *c-fos* gene a relevant component of the activator protein-1 (AP-1) transcription factor, is critical for accurate regulation of numerous genes implicated in cell proliferation, differentiation, apoptosis, and oncogenic transformation [28]. Data presented in this work increases the knowledge regarding BPA-induced effects in the transcriptional response to a frequently utilized therapeutic DOX concentration with potential consequences for chemotherapy outcome and risk assessment.

## 2. Materials and Methods

### 2.1. Cell Cultures and Reagents

In this work, Hep-2 and MRC5 cell lines were kindly offered by Centro Hospitalar Lisboa Ocidental; Hospital Egas Moniz, Microbiology and Molecular Biology Laboratory, and cultivated in 75 cm^2^ flasks with RPMI media containing GlutaMAX™ I, 25 mM HEPES (Invitrogen, Carlsbad, CA, USA). Culture media was supplemented with 10% (*v*/*v*) fetal bovine serum, 100 U/mL penicillin, 100 mg/mL streptomycin and 2 mM L-glutamine. All cell cultures were maintained in a 5% (*v*/*v*) CO_2_ humidified atmosphere at 37 °C. In the subcultures, the cells stabilized for 24 h before treatments.

### 2.2. Drugs and Treatments

Here, BPA (Sigma, St. Louis, MI, USA) was newly diluted in vehicle ethanol and supplemented in the culture media at final concentrations of 4.4 μM (1 µg/mL), 4.4 nM (1 ng/mL) and 0.44 nM (0.1 ng/mL). Relevantly, 0.44 nM dosages, in the range of BPA-detected levels in human biological samples, was associated with environmental exposure, whereas 4.4 μM was selected based on the reference value of 50 μg/kg body weight/day, considering an average body weight of 70 kg and 3 L of water intake per day. Considering that the U.S. Environmental Protection Reference Dose for Chronic Oral BPA Exposure (RfD) is 50 μg/kg body weight/day [29], which is in agreement with the BPA Tolerable Daily Intake (TDI), which is the value set until January 2015 by the European Food Safety Authority, and reduced to 4 μg/kg body weight/day [30] thereafter. For interactions assessment, DOX (AppliChem, Darmstadt, Germany) was diluted in water and supplemented the culture medium at 4 μM (2.5 µg/mL), in final concentration, which correspond to free DOX levels in blood in cancer chemotherapy patients [31]. Regarding, BPA/DOX co-exposures, the cells were previously exposed to BPA for 24 h. Afterwards, 24 h of co-exposure to BPA and DOX followed. Additionally, cell cultures were exposed to BPA for 48 h after the 24 h stabilization period, and DOX standard medium was substituted by the medium with DOX 48 h after subculture for single-drug treatments. In all experiments, the controls were utilized in cells grown in standard culture medium or in medium supplemented with ethanol 170 μM (vehicle concentration for BPA).

### 2.3. RNA Isolation and Real-Time PCR

The total RNA was extracted from cell lysates using the GF-1 Total RNA Extraction Kit (Vivantis technologies, Subang Jaya, Malaysia), according to the manufacturer’s protocol. The concentrations of all RNA samples were determined by a fluorescence-based assay, the Qubit™ RNA HS Assay Kit in Qubit™ 3.0 Fluorometer (Invitrogen, Carlsbad, CA, USA). Then, 1 µg of total RNA was reverse transcribed to cDNA using random hexamers from the First-Strand cDNA Synthesis Kit (GE Healthcare, Little Chalfont, UK). Quantitative real-time PCR (qRT-PCR) was conducted on a CFX Connect™ Real-Time PCR Detection System (Bio-rad) for the genes *c-fos*, *p21*, *bcl-xl* and for both reference genes *GAPDH* and *β-actin*, which were used for data normalization. Each reaction was performed in triplicate using the 5x HOT FIREPol^®^ EvaGreen^®^ qPCR Supermix (Solis BioDyne, Tartu, Estonia) in a final volume of 20 µl. Control PCRs were also performed for all primer combinations without a template. The primers used are listed in Table 1 and the cycling conditions were as follows: Initial activation of 95 °C for 12 min, followed by 40 cycles of 95 °C for 30 s, 55 °C for 30 s, and 72 °C for 40 s. Then, those cycles were followed by the acquisition of a melting curve, in order to check for primer-dimer formation and contaminations. The relative quantification was undertaken by normalizing threshold cycles (Ct) of the target genes with the mean Ct of *GAPDH* and *β-actin*, since no significant differences were detected between these two reference genes. Transcript levels were analyzed by calculating ΔΔCt (ΔΔCt = ΔCt treatment − mean ΔCt control) and the control samples always had a value of 0.

### 2.4. Statistical Analysis

All statistical analysis was performed by using IBM SPSS statistics 22 software (Armonk, NY, USA). Significant differences between different groups were assessed through Student’s *t*-test (comparison for two groups) and *p* < 0.05 was considered statistically significant. Results are presented as mean ± standard deviation.

## 3. Results

### 3.1. BPA Alters bcl-xl and p21 Transcription Levels but Does Not Significantly Affect c-fos Expression

Relative expression of cancer-related genes, namely *bcl-xl*, *c-fos* and *p21,* was analyzed in Hep-2 and MRC-5 cells, through qRT-PCR, after 48 h of exposure to different BPA environmentally relevant concentrations (0.44 nM, 4.4 nM, and 4.4 µM). We observed a pronounced divergence in cellular response between the two tested cell lines. In Hep-2 cells, the higher BPA concentration assessed induced an upregulation (*p* = 0.003) of *bcl-xl* mRNA levels (Figure 1A). No significant changes were observed for *c-fos* mRNA levels after BPA exposure (Figure 1B) and *p21* mRNA expression was upregulated after exposure to 4.4 nM and 4.4 µM BPA concentrations (*p* = 0.018) (Figure 1C). Regarding MRC-5 cell line, no significant differences were observed in *bcl-xl* mRNA expression (Figure 1D) or *c-fos* mRNA expression (Figure 1E). On the other hand, we observed an upregulation of *p21* mRNA levels after exposure to the lowest BPA concentration (*p* = 0.035) (Figure 1F).

### 3.2. BPA Interacts with DOX Transcriptional Effects in Hep-2 and MRC-5 Cell Lines with Cell Type Specificity

The DOX effects alone, and combined with BPA, were assessed using the same primers described above (Figure 2). Ours results demonstrate that DOX severely decreases *bcl-xl*, *c-fos* and *p21* expression levels in Hep-2 cells, with higher significance when compared to vehicle (*p* < 0.001 for all the three genes). However, the combined exposure with BPA antagonized DOX transcriptional effects with the exception of DOX/BPA 0.44 nM combination in *bcl-xl* and *p21* expression (Figure 2A,C). Regarding MRC-5 cells effects, DOX alone did not significantly affect *c-fos* and *p21* expression levels. However, the combined exposure of DOX/BPA resulted in a significant downregulation, and upregulation, respectively of these genes. The anti-apoptotic *bcl-xl*, BPA interaction with DOX were not observed. (Figure 2D). Moreover, the combined effects of DOX/BPA were also dose-dependent in Hep-2 cells namely in *bcl-xl* (DOX/BPA 0.44 nM and DOX/BPA 4.4 nM (*p* = 0.028); DOX/BPA 4.4 nM, DOX/BPA 4.4 µM (*p* = 0.015)) (Figure 2A), and in *p21* expression levels (DOX/BPA 0.44 nM and DOX/BPA 4.4 nM (*p* = 0.009)) (Figure 2C).

## 4. Discussion

The continuous growth of chemo-resistance and the concurrent worldwide increase of man-made products to which human are continuously and ubiquitous exposed has led to the question: Are humans exposed to environmental contaminants that may enhance and/or endorse cellular resistance to chemotherapy?

One of the most utilized drugs in chemotherapy worldwide is DOX, with distinct concentration-dependent cellular effects. DOX often used therapeutic dosages to induce cellular death through cell cycle arrest and apoptosis [32]. Previous studies, in HT29 cell line, demonstrated that the endocrine disruptor, BPA, was able to interfere with DOX transcriptional effects on genes related to cell cycle progression, mitotic regulation and apoptosis [22]. Moreover, considering that, several studies indicated that BPA effects are cell type and dose dependent, here we aimed to evaluate in Hep-2 and MRC-5 cell lines. Environmentally-relevant concentrations of BPA were able to interfere and/or antagonize DOX cellular effects in transcription levels of *c-fos*, *p21* and *bcl-xl*, as previously reported in HT29 cell line.

Cellular cytotoxicity and genotoxicity of BPA tested concentrations, namely, 4.4 μM (1 µg/mL), 4.4 nM (1 ng/mL), and 0.44 nM (0.1 ng/mL), which correspond to the range of BPA detected levels in human biological samples due to environmental exposure, have already been evaluated in Hep-2 and MRC-5 cell lines in a previously published study [26]. We have demonstrated that these BPA concentrations do endorse genotoxic effects, such as micronuclei formation at 4.4 nM and 0.44 nM concentrations and DNA, as well as oxidative damage with no detected effects in cellular viability, with associated non-monotonic responses [26].

One of the generally recognized genes acting as a key player in cancer biology is the anti-apoptotic *bcl-xl* [33]. We have previously demonstrated that in HT29 cell line BPA at 4 uM induced a significant transcriptional upregulation [22], as was also observed in this study in Hep-2 cell line. Moreover, La Pensee and coworkers have correlated BPA antagonistic effect to DOX with increased expression of anti-apoptotic proteins [20]. Here we also demonstrate that BPA induces an antagonistic effect of DOX transcriptional effect in Hep-2 cells, and more importantly, this effect is particularly observed within DOX interactions, with the lowest tested BPA concentration, which indicates a non-monotonic response effect. On the other hand, in MCR-5 cells, no significant effects were observed after BPA exposure alone, or in co-exposure with DOX, which may indicate that transcriptional alterations in the expression levels of this anti-apoptotic gene are cell type-specific and are more likely to appear in tumors cells. Considering that overexpression of *bcl-xl* impairs apoptosis, induced by DOX [33] and that DOX therapeutic concentrations induce apoptosis [32], BPA antagonizing effect may negatively influence DOX therapeutic outcomes.

Moreover, another key gene in cancer biology is *c-fos*, which encodes a crucial component of the activator protein-1 (AP-1) transcription factor, required for accurate regulation of several genes associated with cell proliferation, differentiation, apoptosis, and oncogenic transformation [28]. In agreement with previous studies [22], *c-fos* transcriptional levels were not affected by BPA exposure; however, the interaction with DOX was obvious for both cell lines. Remarkably, although BPA interaction occurs in both cells the effects are divergent. While, in Hep-2 cells BPA decreases the downregulation induced by DOX exposure, which is concordant with previous results in HT29 cancer cell line [22], in MRC-5 cells BPA enhances *c-fos* downregulation with more pronounced effects in the lowest tested concentrations.

The third analyzed gene, *p21* encodes a cyclin-dependent kinase inhibitor is tightly connected with cell cycle progression and cancer outcome [27]. BPA effect in the transcriptional expression of this gene emphasizes once more its cellular specificity. In Hep-2 cells, we observed an upregulation of transcriptional levels at the highest tested concentrations, which indicates a dose-dependent effect, while in MRC-5 cells, *p21* upregulation was observed at the lowest BPA concentration, which demonstrates a non-monotonic effect. Interestingly, the BPA interaction with DOX is evident and results in striking divergent effects. Whereas, in Hep-2, BPA antagonizes DOX induced downregulation of *p21* in a dose-dependent manner, while in MRC-5 cells BPA co-exposure with DOX results in a very significant upregulation of gene expression. Interestingly, previous studies in colon epithelial cells have also demonstrated an inversed correlation between *c-fos* and *p21* expression associated with oxidative stress, ref [34] which we have demonstrated occurs in these cells (Hep-2 and MRC-5) after BPA and DOX co-exposure [26].

Overall our data demonstrate that Hep-2 cells were more susceptible to BPA effects in a dose-dependent manner, while MRC-5 transcriptional levels present non-monotonic response. Interestingly, previous studies demonstrate that the tested BPA concentrations induced higher levels of DNA damage and oxidative stress in Hep-2 cells, rather than MRC-5 also in a dose-dependent manner [26].

## 5. Conclusions

In this study, we provide new evidence regarding BPA cell type-specific effects and non-monotonic responses. Furthermore, we demonstrate that relevant environmental doses of BPA directly affect transcriptional levels of key player genes in cancer biology and interact with DOX-induced effects, that might be crucial for cancer patients undergoing this treatment. Our data emphasize the necessity of a better understanding of BPA interactions with chemotherapeutic agents in the context of risk assessment.

## Figures and Tables

**Figure 1 toxics-07-00043-f001:**
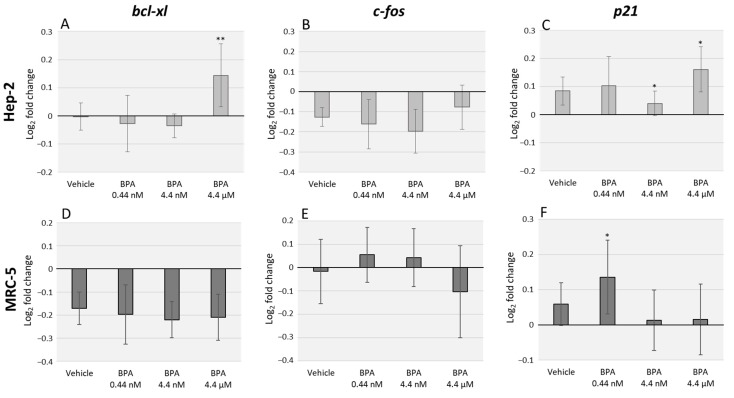
Quantitative real-time PCR (qRT-PCR) analysis of Hep-2 (**A**–**C**) and MRC-5 (**D**–**F**) cell lines after exposure to three different concentrations of Bisphenol A (BPA). The data represent the relative expression of the genes: *bcl-xl* (**A**,**D**), *c-fos* (**B**,**E**), and *p21* (**C**,**F**). *GAPDH* and *β-actin* were used for normalization. The error bars represent the standard deviation between two independent treatments and three qRT-PCR replicates. Significant statistical values, which were compared with the vehicle (EtOH) and calculated with Student’s *t*-test, are illustrated as: * *p* < 0.05 and ** *p* < 0.01.

**Figure 2 toxics-07-00043-f002:**
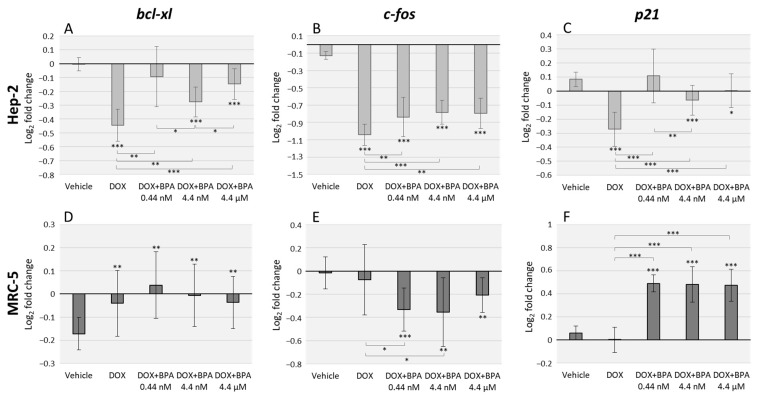
qRT-PCR analysis in Hep-2 (**A**–**C**) and MRC-5 (**D**–**F**) cell lines after exposure to doxorubicin (DOX) alone and in combination with three different environmentally relevant concentrations of BPA namely 0.44 nM, 4.4 nM, and 4.4 µM. Data represent the relative expression for the genes: *bcl-xl* (**A**,**D**), *c-fos* (**B**,**E**) and *p21* (**C**,**F**). *GAPDH* and *β-actin* were used for normalization. Error bars represent the standard deviation between two independent treatments and three qRT-PCR replicates. Significant statistical values, which were compared with the vehicle (EtOH) and calculated with Student’s *t*-test, are illustrated as: * *p* < 0.05, ** *p* < 0.01 and *** *p* < 0.001.

**Table 1 toxics-07-00043-t001:** Primer sequences, accession numbers, and product lengths for quantitative real-time PCR (qRT-PCR) analysis.

Genes	Accession Number *	Forward Primer (5′→3′)	Reverse Primer (3′→5′)	Product Length (bp)
***GAPDH***	NM_002046	GAGTCAACGGATTTGGTCGTA	GCAGAGATGATGACCCTTTTG	245
***β-actin***	NM_001101	AGGCCAACCGCGAGAAG	ACAGCCTGGATAGCAACGTACA	79
***bcl-xl***	Z23115.1	TTACCTGAATGACCACCTA	ATTTCCGACTGAAGAGTGA	185
***c-fos***	NM_005252	AGGAGAATCCGAAGGGAAAG	CAAGGGAAGCCACAGACATC	247
***p21***	NM_000389	CTGGAGACTCTCAGGGTCGAA	CCAGGACTGCAGGCTTCCT	123

* GenBank accession numbers (National Center for Biotechnology).

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
