# Peer review of "Environmentally Relevant Concentrations of Bisphenol A Interact with Doxorubicin Transcriptional Effects in Human Cell Lines"

_toxics, 2019, doi:10.3390/toxics7030043_

Round 1

Reviewer 1 Report

The introduction seems to be a little short, but it does cover the main topics. The introduction could benefit from a bit more detail further describing BPA, EDCs and DOX. 

The design of the experiment seems to be solid, and the conclusions are reasonable. Obviously, more research needs to be performed to completely explain these results, but this article is a solid first step. 

The only real problem is the level of English in the article. There are many many grammatical errors, and mis-spellings (booth instead of both or trough instead of through). I suggest that you have a good native speaker proofread the manuscript. Once the language is corrected, this will be a solid article about a subject that we need more published information about.     

Author Response

We were very pleased with the reviewer feedback and grateful for the pertinent recommendations. We have improved the introduction section of the manuscript with the introduction of more details regarding BPA analogues and derivates and interactions with key players in cancer biology.

We have also addressed the language correction and performed an extensive English revision with an appropriate software.  

Reviewer 2 Report

This manuscript submitted by Edna Ribeiro and coworkers describes findings that enviromentallyrelevant concentrations of Bisphenol A interact with Doxorubicin-induced effects in human cell lines.

Overall, this is a concise paper that reports on the interference of the endocrine disruptor chemical Bisphenol A and therapeutic drugs such as doxorubicin, which might be of importance for chemotherapy.

This paper is acceptable after the following minor revisions have been carried out:

The introduction could be expanded and improved by incroporating and citing additional, recently published papers that report on broader effects of Bisphenol A such as:

1) Seachrist et al. (2016) Reproductive Toxicology 56, 167-82

2) Chen et al. (2016) Environ. Sci. Technol. 50, 11, 5438-5453

3) Schoepel et al. (2013) J. Med. Chem. 56, 23, 9664-9672

4) Schoepel et al. (2018) Int. J. Mol. Sci. 19(4), 1133

Figures 1 and 2 should be revised as follows:

a) the axes labels need to be enlarged to allow for better readability

b) the resolution should be increased

The reference list should be checked and, if necessary, corrected to remove special characters:

reference no. 6: check special characters and questions marks following the wor receptor

reference no. 15: check spelling of receptor-??-positive

Finally, the spelling and layout of the text should be carefully checked again to correct minor errors such as:

line 48: remove spaces between respectively/6/,

line 50: remove spaces between pathway/8-11

line 54: remove space between chemoresistance/14

line 155: correct as follows: does not significantly affect the

line 156: correct to: difference between these

line 160/161: correct to: The compounds were also tested in combination

line 210: remove space between transformation/25

Author Response

We were extremely pleased with the reviewer feedback and acknowledged the reviewer most relevant suggestions.

We have altered the manuscript accordingly with the reviewer constructive suggestions regarding the improvement of the introduction section and included in the manuscript suggested references, namely:

2) Chen et al. (2016) Environ. Sci. Technol. 50, 11, 5438-5453

3) Schoepel et al. (2013) J. Med. Chem. 56, 23, 9664-9672

4) Schoepel et al. (2018) Int. J. Mol. Sci. 19(4), 1133

Apropos to the reviewer suggestions regarding Figures 1 and 2, we have improved and enlarged the axes labels for suitable readability.

Regarding, the reviewer comment of the language correction spelling and layout, we performed an extensive English revision with appropriate software and the reference list checked and revised.